# 2.8 Å resolution reconstruction of the *Thermoplasma acidophilum* 20S proteasome using cryo-electron microscopy

Melody G Campbell[1,2,†], David Veesler[1,2,3,†], Anchi Cheng[1,2,4], Clinton S Potter[1,2,4], Bridget Carragher[1,2,4]*

[1]National Resource for Automated Molecular Microscopy, The Scripps Research Institute, La Jolla, United States; [2]Department of Integrative Structural and Computational Biology, The Scripps Research Institute, La Jolla, United States; [3]Department of Biochemistry, University of Washington, Seattle, United States; [4]New York Structural Biology Center, New York, United States

**Abstract** Recent developments in detector hardware and image-processing software have revolutionized single particle cryo-electron microscopy (cryoEM) and led to a wave of near-atomic resolution (typically ∼3.3 Å) reconstructions. Reaching resolutions higher than 3 Å is a prerequisite for structure-based drug design and for cryoEM to become widely interesting to pharmaceutical industries. We report here the structure of the 700 kDa *Thermoplasma acidophilum* 20S proteasome (T20S), determined at 2.8 Å resolution by single-particle cryoEM. The quality of the reconstruction enables identifying the rotameric conformation adopted by some amino-acid side chains (rotamers) and resolving ordered water molecules, in agreement with the expectations for crystal structures at similar resolutions. The results described in this manuscript demonstrate that single particle cryoEM is capable of competing with X-ray crystallography for determination of protein structures of suitable quality for rational drug design.

*For correspondence: bcarr@nysbc.org

[†]These authors contributed equally to this work

Competing interests: The authors declare that no competing interests exist.

## Introduction

Single-particle cryo-electron microscopy (cryoEM) is currently undergoing a revolution due to the recent development of a new generation of detectors using the complementary metal-oxide semiconductors (CMOS) technology (*Kuhlbrandt, 2014*). These cameras directly detect incoming electrons without the need for a scintillator converting the electrons into photons and are characterized by improved detective quantum efficiencies at all spatial frequencies compared to traditional charge-coupled device cameras (CCDs) or photographic films (*Ruskin et al., 2013*; *McMullan et al., 2014*). The fast read-out rate of these devices also allows for recording movies composed of multiple frames during typical image exposure times enabling correction of beam-induced sample motion and stage drift to reduce image blurring (*Brilot et al., 2012*; *Shigematsu and Sigworth, 2013*).

Several algorithms have been developed for tracking beam-induced motion and stage drift in movies recorded with direct detectors (*Campbell et al., 2012*; *Bai et al., 2013*; *Li et al., 2013*). These data processing strategies have been used to produce a multitude of high-resolution reconstructions of samples in the MDa mass range (*Amunts et al., 2014*; *Fernandez et al., 2014*; *Voorhees et al., 2014*; *Wong et al., 2014*; *Allegretti et al., 2014*) as well as of protein complexes once considered too small for single-particle cryoEM (≤300 kDa) (*Cao et al., 2013*; *Liao et al., 2013*; *Lu et al., 2014*). These achievements constitute a tremendous leap forward for single-particle cryoEM. This method

**eLife digest** Proteins perform many critical tasks within cells, and to do so, they must first fold into specific shapes. Being able to visualize these shapes can help scientists to understand how proteins work, and help them create drugs that can interact with the proteins to treat diseases.

The past few years have seen the rapid development of an imaging technique called single-particle cryo-electron microscopy (or cryoEM for short), and this technique is now increasingly used to investigate protein structures. First, proteins are embedded in a thin film of non-crystalline ice by rapidly cooling to around the temperature of liquid nitrogen (below −180°C). This traps the protein in the shape it has in solution. High-energy electrons are then transmitted through the protein sample and their interaction with the atoms in the protein is recorded by a direct electron camera. The analysis of a large series of images recorded in this way can be used to determine the approximate positions of the atoms in the protein.

Previously, single-particle cryoEM techniques have not produced a detailed enough protein structure to be useful to scientists interested in drug development. By refining these techniques, Campbell, Veesler et al. have now obtained the most detailed cryoEM protein structure to date—a structure of an enzyme complex that helps get rid of proteins that are misfolded or that have become too abundant. The structure is so detailed that it reveals the shapes of some small groups of atoms that stick out from the sides of amino acids in the enzyme complex. (Amino acids are the building blocks of enzymes and all other proteins.) Moreover, the structure shows where individual water molecules are positioned around the protein.

The level of detail in the structure produced by Campbell, Veesler et al. is high enough to be useful to drug researchers. Furthermore, because only 10% of the images Campbell, Veesler et al. collected were used to produce the structure, future work will investigate whether incorporating more of the images could reveal structures in even greater detail.

can now compete with X-ray crystallography for determination of protein structures at near-atomic resolution while also offering the unique advantage of enabling the characterization of heterogeneous or flexible protein complexes (*Scheres et al., 2007*). High-resolution cryoEM can thus provide information about protein dynamics, which is key to understanding the functions of many macromolecular complexes.

We report here the structure of the *Thermoplasma acidophilum* 20S proteasome (T20S), determined at 2.8 Å resolution by single-particle cryoEM. This enzyme is a key component of the cell metabolism carrying out degradation of ubiquitinylated polypeptides to regulate the concentration of specific proteins as well as eliminating misfolded products. The T20S proteasome forms a 700 kDa complex comprising 14 α and 14 β subunits organized with D7 symmetry. Our reconstruction enables identification of the conformational preference of some amino-acid side chains (rotamers) and resolves ordered water molecules, in agreement with the expectations for crystal structures at similar resolutions. These outcomes constitute a significant step forward in overcoming the resolution limit achievable by single particle cryoEM, which can now provide protein structures of suitable quality for rational drug design.

## Results and discussion

Careful alignment of the microscope was performed before data collection to ensure Thon rings were visible up to ~2.5 Å resolution in the power spectrum of micrographs collected over amorphous carbon using the same conditions as for data acquisition. Prior to beginning data collection, coma-free alignment was carried out to align the beam to the column optical axis with the assistance of a Zemlin tableau (*Glaeser et al., 2011*). Automated data collection was carried out using Leginon (*Suloway et al., 2005*) to control both the FEI Titan Krios microscope and the Gatan K2 Summit camera operated in 'super-resolution' mode. A subset of micrographs was automatically selected using a filter built into the Appion pipeline (*Lander et al., 2009*) that accepts images based on the resolution limit to which Thon rings can be confidently identified in the image power spectra. This is achieved by computing cross-correlation coefficients (CC) between the 1-D radially averaged power spectrum of each micrograph and the calculated Contrast Transfer Function (CTF). Only those micrographs showing a CC ≥ 80% at a resolution of 4 Å or better were retained for further processing. One of these micrographs and its corresponding power spectrum are shown in *Figure 1*. Using this selection

criterion, 196 micrographs were selected, from which a total of 87,066 particles were automatically picked using FindEM (*Roseman, 2004*). These movies have been deposited in the EMPIAR database with the accession number EMPIAR-10025. Particle images were sorted and selected based on mean and standard deviation pixel values within Appion (*Lander et al., 2009*) and using Xmipp cl2d reference-free alignment and clustering (*Sorzano et al., 2010*) to yield a stack of 59,864 particles containing both top and side views that were then used for refinement and reconstruction.

The selected substack of T20S particles was refined starting from a low-pass filtered initial model to high resolution using the gold standard procedure implemented in Relion (*Scheres and Chen, 2012*; *Scheres, 2012b*; *Bai et al., 2013*). This projection-matching refinement step provided a reconstruction at 2.98 Å resolution (estimated angular accuracy 1.3°) using all 38 movie frames and the full exposure of 53 e⁻/Å². Previous measurements using catalase crystals have suggested that exposures as low as 11 e⁻/Å² might be necessary to attain 3 Å maps (*Baker and Rubinstein, 2010*; *Baker et al., 2010*). However, the loss of signal due to radiation must be balanced against the need for sufficient contrast in the images to allow for accurate alignment of the individual particles. Collecting data in movie mode using direct detectors has profoundly changed the way data is acquired and processed: many reported reconstructions have been obtained after eliminating sub-optimal frames (*Li et al., 2013*) or applying a frequency-dependent weighting scheme to the frames (*Scheres, 2014*). We emphasize that we were able to obtain a T20S reconstruction at 2.98 Å resolution using the full exposure of 53 e⁻/Å² and prior to weighting individual movie frames. We obtained similar results in determining the structure of an icosahedral virus to 3.7 Å resolution using an exposure of 38 e⁻/Å² with a microscope operated at 200 kV (*Campbell et al., 2014*). These results contrast with previous measurements and demonstrate that useful high-resolution information may survive much higher exposures than previously thought for single particle cryoEM.

We subsequently used the particle polishing procedure implemented in version 1.3 of the Relion software to account for individual beam-induced particle translations and to calculate a frequency-dependent weight for the contribution of individual movie frames to the reconstruction (*Scheres, 2014*). Plots of the relative B factor and $C_f$ intersect, derived from the Guinier plots for each single frame reconstruction, along with the corresponding frequency-dependent weights are presented in *Figure 2*. The first frame of each movie featured a large movement in comparison with the following ones, which we and others interpret as the initial settling of beam-induced motion during exposure. This processing scheme improved the resolution of the reconstruction to 2.83 Å (estimated angular accuracy 1.0°), and this was supported by a significant enhancement of map quality.

In a final round of refinement, we discarded particle images with the greatest angular uncertainty based on the height of the probability distributions at their maximum. An empirically determined cutoff of 0.07 was used, which gave a stack of 49,954 particle images (699,356 asymmetric units).

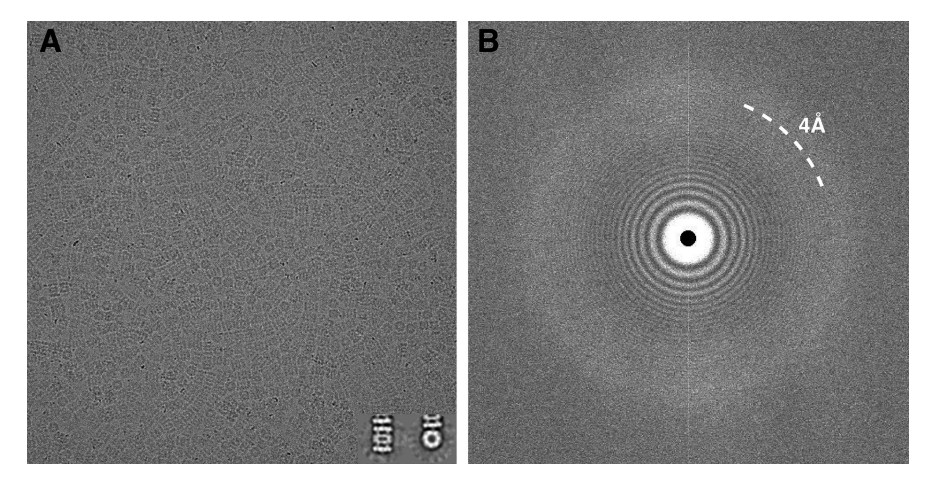

**Figure 1**. Typical micrograph of ice-embedded T20S proteasome and corresponding power spectrum. (**A**) Micrograph of T20S after movie-frame alignment. Inset: Reference free 2D class averages showing a side view (left) and a top view (right). (**B**) Thon rings are visible well beyond 4 Å⁻¹ resolution in the power spectrum of the micrograph shown in (**A**). DOI: 10.7554/eLife.06380.003

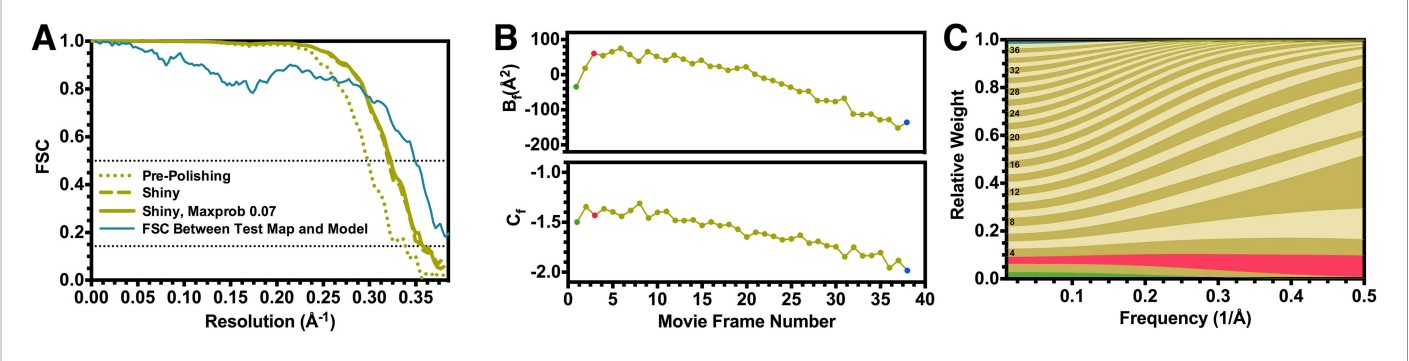

**Figure 2**. Fourier shell correlation curves and radiation damage weighting plots. (**A**) Gold-standard FSC curves for the T20S reconstructions before and after particle polishing as well as after excluding particles based on the uncertainty of the angular assignments. Estimated resolutions are 2.98 Å, 2.83 Å, and 2.81 Å, respectively. The FSC curve computed between the atomic model and the test map is shown in blue. The FSC reaches 0.5 at 2.86 Å resolution, in agreement with the resolution estimated by gold standard refinement in Relion. (**B**) Estimated values for $B_f$ (top) and $C_f$ (bottom) during the particle polishing procedure. (**C**) Frequency-dependent relative weights for all movie frames. The first, third, and final movie frames are highlighted in green, red, and blue, respectively.

Refinement of this smaller dataset under otherwise identical conditions produced a map with a slightly improved resolution of 2.81 Å (estimated angular accuracy 0.8°). This result suggests that analyzing the probability distributions over all orientations is a convenient way of discarding particles that are not contributing positively to the reconstruction (~10,000 particles in this case). This map has been deposited in the EMDB with the accession number EMD-6287.

We fitted the T20S atomic coordinates obtained from a crystal structure at 1.9 Å resolution (*Forster et al., 2005*) into our cryoEM reconstruction. The density is continuous from residues 13 to

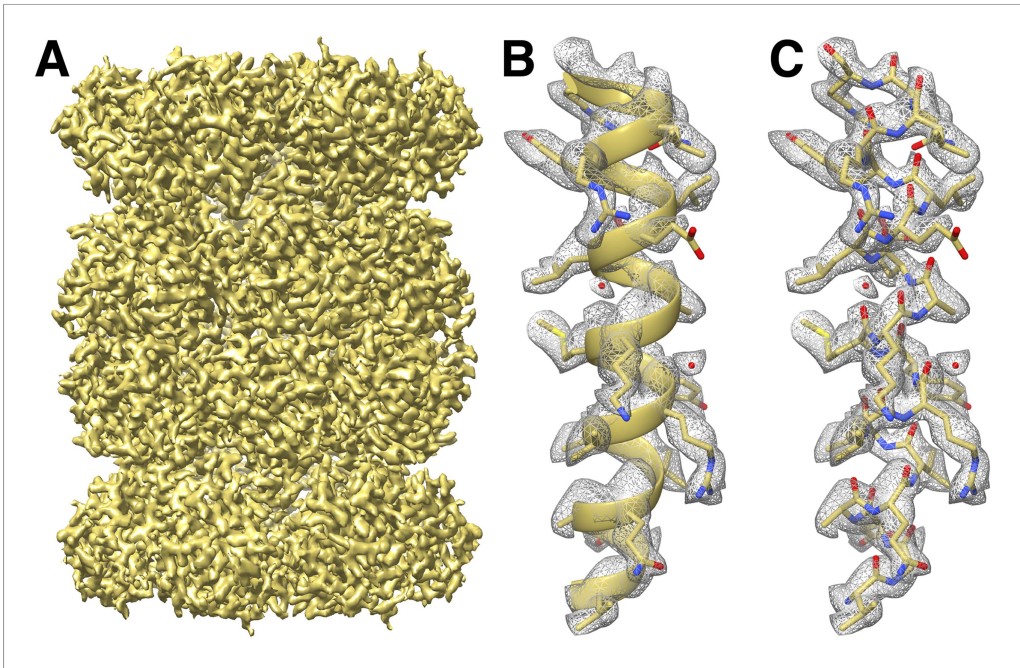

**Figure 3**. CryoEM reconstruction of the T20S at 2.8 Å resolution. (**A**) Isosurface representation of the T20S map. (**B**) An α-helical segment from one β subunit is shown in ribbon representation docked into the corresponding region of the reconstruction. (**C**) Same α-helical segment as in (**B**) shown in atom representation docked into the corresponding region of the reconstruction. Several water molecules are visible.

233 and residues 1 to 203 for α and β subunits, respectively, with most side chains being resolved (*Figure 3*). In line with previous studies (*Allegretti et al., 2014*; *Bartesaghi et al., 2014*), many negatively charged residues had lost their carboxylate groups, likely due to radiation sensitivity. Our T20S reconstruction shows many features supporting the resolution claim of 2.8 Å. Differences between the crystal structure and the cryoEM reconstruction are easily discernable at the level of the backbone and can be adjusted accordingly using real-space refinement procedures. The rotameric conformations of many amino-acid side chains are clearly visible in the electron potential map (*Figure 4*) and reveal differences with a crystal structure of the T20S determined at 3.4 Å resolution (*Lowe et al., 1995*). The quality of our reconstruction allows distinguishing between Phe and Tyr amino-acid residues based on the appearance of the density for their side chains (*Figure 4*), as shown previously (*Zhang et al., 2010*). The presence of a hydroxyl group para to the phenyl group of Tyr residues unambiguously establishes the presence of the Tyr residue as opposed to the Phe residue in our reconstruction. This level of resolvability is decisive for de novo building of atomic models of proteins of unknown structures.

In X-ray crystallography, when a structure is determined at a resolution significantly better than 3 Å, one expects to observe ordered water molecules interacting with amino-acid residues located at the protein surface (*Carugo and Bordo, 1999*). The higher the resolution the more ordered water molecules are visible. Analysis of our T20S reconstruction at 2.8 Å resolution revealed that many

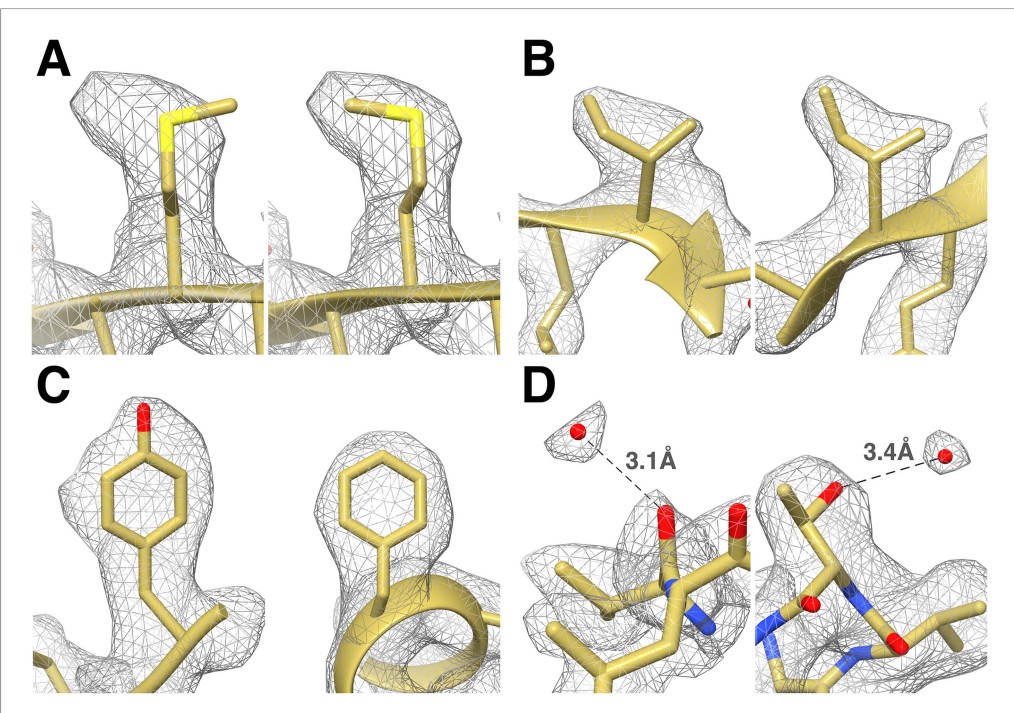

**Figure 4**. Identification of the rotameric conformation of amino acid side chains and resolving of ordered water molecules in the T20S cryoEM reconstruction at 2.8 Å resolution. (**A**) Different rotameric conformations adopted by the Met-14 side chain (β-subunit) between a crystal structure of the T20S determined at 3.4 Å resolution (left, PDB 1PMA) and the EM density (right). (**B**) Unambiguous establishment of the rotameric conformations of two different isoleucine residues: Ile70 (left, α-subunit) and Ile37 (right, β-subunit). (**C**) The additional density accounting for the hydroxyl group of tyrosine side chains (left, Tyr132, α-subunit) is prominent when compared to phenylalanine side chains (right, Phe91, α-subunit). (**D**) Numerous water molecules are resolved in the T20S cryoEM map. Left: a water molecule hydrogen-bonded to the carbonyl group of Val87 (α-subunit). Right: a water molecule hydrogen-bonded to the side chain hydroxyl of Thr102 (α-subunit). The cross-validation of the assignment of these water molecules using gold-standard half maps can be found in *Figure 4—figure supplement 1*.

The following figure supplement is available for figure 4:

**Figure supplement 1**. Cross-validation of water molecule assignment through comparison of gold standard half maps.

ordered water molecules are resolved, in agreement with the estimated resolution. They are positioned in appropriate chemical environments and within hydrogen-bonding distance (2.8–3.5 Å) of the surrounding atoms with which they interact. All the reconstructions described in this manuscript used the RELION gold standard refinement procedure (*Scheres and Chen, 2012*). In this protocol, two half datasets are refined completely independently throughout, and after each cycle the new reference volumes are low-pass filtered to the resolution where the Fourier shell correlation (FSC) between the two volumes drops to 0.143 (*Rosenthal and Henderson, 2003*), thus preventing overfitting of noise. As a result, the noise present in each independent map is not correlated. We were therefore able to cross-validate our assignment of water molecules as they are accounted for by the density in both maps derived from half datasets despite the reduced number of particle images contributing to each map (*Figure 4—figure supplement 1*) (*DiMaio et al., 2013*). Finally, the 1.9 Å crystal structure of the T20S proteasome also unequivocally supports the assignment of the water molecules identified in the cryoEM map as an independent cross-validation metric (*Forster et al., 2005*).

The structure presented here demonstrates that cryoEM is now capable of producing structures at better than 3 Å resolution, which is a prerequisite for structure-based drug design (*Anderson, 2003*). The possibility to resolve the conformational preference of some amino-acid side chains (rotamers) and identify ordered water molecules opens new horizons for cryoEM and structural biology in general. This level of information is required for drug lead design as these chemical entities mediate key biological processes such as catalysis, protein/protein and protein/substrate recognition. State-of-the-art cryoEM is now in a position to expedite structure-based drug discovery and will likley become increasingly important for pharmaceutical industries.

We believe several factors contributed to obtaining the resolution reported here. The proteasome itself is a rigid and homogenous sample, which is ideal for high-resolution reconstruction. For data collection, we used a relatively high electron dose, which may have been key to properly aligning particles. Furthermore, we used the mechanical stage instead of beam-tilt to move to targets for high magnification exposure in order to avoid introducing any phase shift in the images. Images of each square and each hole were examined by-eye for perceived ice thickness and only those which were judged to have very thin ice were selected for final high magnification exposures. Finally, we benefited from the availability of algorithmic advances (projection matching, particle polishing), which allows optimization of the doses used for alignment and final reconstruction.

As the potential of cryoEM has been steadily growing in the past few years, the prospect of achieving reconstructions at ~2 Å resolution is very attractive. Remarkably, we used only 10% of the particle images present in the dataset discussed in this manuscript. Including more particle images could further improve the resolution of the T20S reconstruction and experiments are ongoing to determine if this is the case. Finally, 3D classification approaches have also proven useful for sorting particle images (*Scheres, 2012a*; *Lyumkis et al., 2013*) and implementing such approaches may also help to improve the final resolution of our reconstruction.

## Materials and methods

### Data collection

The *T. acidophilum* 20S proteasome was expressed and purified from *Escherichia coli* according to established protocols (*Rabl et al., 2008*; *Yu et al., 2010*). Three microliters of sample were applied to a 1.2/1.3 C-flat grid (Protochips, Raleigh, North Carolina), which had been plasma-cleaned for 6 s at 20 mA using a Gatan Solarus (Pleasanton, California). Thereafter, grids were plunge-frozen in liquid ethane using a Gatan CP3 and a blotting time of 2.5 s. Data were acquired using an FEI Titan Krios (Hillsboro, Oregon) transmission electron microscope operated at 300 kV and equipped with a Gatan K2 Summit direct detector. The extraction voltage was 4500, the gun lens setting 3 and the spotsize 8. A condenser aperture of 70 µm and an objective aperture of 100 µm were used. Coma-free alignment was performed using the Leginon software (*Glaeser et al., 2011*). Automated data collection was carried out using Leginon (*Suloway et al., 2005*) to control both the FEI Titan Krios (used in microprobe mode at a nominal magnification of 22,500×) and the Gatan K2 Summit operated in 'super-resolution' mode (super-resolution pixel size: 0.6575 Å) at a dose rate of ~9 counts/physical pixel/s which corresponds to ~12 electrons/physical pixel/s (when accounting for coincidence loss). We waited 40 s after physically moving the stage to each new position to allow settling before acquiring a new video. A single movie was taken per hole. Each movie had a total accumulated

exposure of 53 e⁻/Å² fractionated into 38 frames of 200 ms (yielding movies of 7.6 s duration). A dataset of ~1000 micrographs was acquired for the T20S in a single session using a defocus range between 0.9 and 2.4 μm.

## Data processing

Whole frame alignment was carried out using the software developed by *Li et al. (2013)*, which is integrated into the Appion pipeline (*Lander et al., 2009*), to account for stage drift and beam-induced motion. We used a frame offset of seven along with a B factor of 1000 pixels² for aligning the movie frames (*Li et al., 2013*). The parameters of the microscope contrast transfer function were estimated for each micrograph using ctffind3 (*Mindell and Grigorieff, 2003*). Particles were automatically picked using FindEM (*Roseman, 2004*) integrated into the Appion pipeline (*Lander et al., 2009*) before sorting and selection based on mean and standard deviation pixel values and using Xmipp cl2d reference-free alignment and clustering (*Sorzano et al., 2010*). Extraction of particle images for projection-matching refinements was performed using Relion 1.3 with an initial box size of 448 pixels² and applying a windowing operation in Fourier space to downsize the particles images to yield a final box size of 300 pixels² (corresponding to a pixel size of 0.98 Å). The final pixel size was chosen to optimize the balance between the Nyquist frequency limit and the memory requirements for computational steps. Projection-matching refinements were performed with the Relion software (*Bai et al., 2013*; *Scheres, 2012a*, *2012b*, *2014*) imposing D7 symmetry. Reported resolutions are based on the gold-standard FSC = 0.143 criterion (*Scheres and Chen, 2012*) and Fourier shell correction curves were corrected for the effects of soft masking by high-resolution noise substitution (*Chen et al., 2013*).

## Model building

The T20S crystal structure (PDB 1YAR) was rigid body fitted into the cryoEM map using UCSF Chimera (*Goddard et al., 2007*) and then iteratively refined using Rosetta (*DiMaio et al., 2009*) and Coot (*Emsley et al., 2010*). We used Rosetta tools recently developed for cryoEM (*DiMaio et al., 2015*; *Wang et al., 2015*) to perform fragment rebuilding, torsion angles and cartesian minimization of atomic positions and B factor refinement. At each iteration, the best model was visually inspected in Coot and a few amino acid side chain rotamers were manually adjusted to best fit the density. Water molecules were also manually added using Coot after the first iteration of Rosetta refinement. Rosetta and Coot refinements were performed using a training map corresponding to one of the two maps generated by the gold-standard refinement procedure in Relion. The second map (testing map) was used only for calculation of the FSC compared to the atomic model (*DiMaio et al., 2013*). The quality of the final model was analyzed with Molprobity (*Chen et al., 2010*).

## Acknowledgements

This work was supported by a FP7 Marie Curie IOF fellowship (273427) to DV and an American Hearth Association predoctoral fellowship to MGC (14PRE18870036). This research was conducted at the National Resource for Automated Molecular Microscopy which is supported by the NIH and the NIGMS (GM103310). We are grateful to Yifan Cheng and Kiyoshi Egami for kindly providing the T20S sample used in this study. We thank Jean-Christophe Ducom, and the rest of the high-performance computing facility for their assistance. We are also grateful to Frank DiMaio for providing early access to and support for Rosetta tools.

## Additional information

### Funding

| Funder | Grant reference | Author |
| --- | --- | --- |
| National Institute of General Medical Sciences (NIGMS) | GM103310 | Clinton S Potter, Bridget Carragher |
| American Heart Association (AHA) | 14PRE18870036 | Melody G Campbell |
| European Commission Directorate-General for Research and Innovation | Marie Curie IOF fellowship (273427) | David Veesler |

| Funder | Grant reference | Author |
|---|---|---|
| National Institutes of Health (NIH) | GM103310 | Clinton S Potter, Bridget Carragher |

The funders had no role in study design, data collection and interpretation, or the decision to submit the work for publication.

## Author contributions

MGC, DV, Conception and design, Acquisition of data, Analysis and interpretation of data, Drafting or revising the article; AC, Conception and design, Analysis and interpretation of data; CSP, BC, Conception and design, Analysis and interpretation of data, Drafting or revising the article

# Additional files

## Major datasets

The following datasets were generated:

| Author(s) | Year | Dataset title | Dataset ID and/or URL | Database, license, and accessibility information |
|---|---|---|---|---|
| Campbell MG, Veesler D, Cheng A, Potter CS, Carragher B | 2015 | Raw 38-frame movies, used to reconstruct the Thermoplasma acidophilum 20S proteasome to 2.8 Å resolution; Aligned 38-frame movies, used to reconstruct the Thermoplasma acidophilum 20S proteasome to 2.8 Å resolution; Averaged movies, used to reconstruct the Thermoplasma acidophilum 20S proteasome to 2.8 Å resolution | EMPIAR-10025; http://www.ebi.ac.uk/pdbe/emdb/empiar/entry/10025/ | Publicly available at EMPIAR (http://www.ebi.ac.uk/pdbe/emdb/empiar/entry/10025). |
| Campbell MG, Veesler D, Cheng A, Potter CS, Carragher B | 2015 | 2.8 Angstrom resolution reconstruction of the T20S proteasome | EMD-6287; http://www.ebi.ac.uk/pdbe/entry/EMD-6287 | Publicly available at EMDataBank (http://www.ebi.ac.uk/pdbe/entry/EMD-6287). |

The following previously published datasets were used:

| Author(s) | Year | Dataset title | Dataset ID and/or URL | Database, license, and accessibility information |
|---|---|---|---|---|
| Forster A, Masters EI, Whitby FG, Robinson H, Hill CP | 2005 | Structure of Archeabacterial 20S proteasome mutant D9S-PA26 complex | 1YAR; http://www.pdb.org/pdb/explore/explore.do?structureId=1YAR | Publicly available at RCSB Protein Data Bank (http://www.pdb.org/pdb/). |
| Lowe J, Stock D, Jap B, Zwickl P, Baumeister W, Huber R | 1995 | Proteasome from Thermoplasma Acidophilum | 1PMA; http://www.pdb.org/pdb/explore/explore.do?structureId=1PMA | Publicly available at RCSB Protein Data Bank (http://www.pdb.org/pdb/). |

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
