## [Decision Letter]

Thank you for sending your work entitled “2.8 Å resolution reconstruction of the Thermoplasma acidophilum 20 S proteasome using cryo-electron microscopy” for consideration at *eLife*. Your article has been favorably evaluated by John Kuriyan (Senior editor) and 3 reviewers, one of whom is a member of our Board of Reviewing Editors.

The following individuals responsible for the peer review of your submission have agreed to reveal their identity: Sjors Scheres (Reviewing editor) and Hong Zhou (peer reviewer). A third reviewer remains anonymous.

The Reviewing editor and the other reviewers discussed their comments before we reached this decision, and the Reviewing editor has assembled the following comments to help you prepare a revised submission.

Where many other cryo-EM reconstructions have recently been solved at just above 3 Å resolution, the 20S proteasome structure at 2.8 Å resolution in this paper goes significantly beyond that. There was strong consensus among the reviewers that this structure represents another significant step forward in the recent, fast advances in cryo-EM structure determination, and that therefore the paper should be published in *eLife* as soon as possible.

The following two things were discussed that would improve the manuscript.

1) All 3 reviewers mentioned in some form that it would be good if you could elaborate somewhat on the obtained atomic model. Are there any differences in the rotamer conformations between your cryo-EM structure and the 1.9 Å X-ray structure? And at this resolution do you observe similar effects of radiation damage on specific side chain densities as for example in (Allegretti et al., *eLife* 3, e01963, 2014; Bartesaghi et al., PNAS 111, 11709-14, 2014), who have shown that the majority of carboxylate chains are missing in cryo-EM maps, supposedly due to their high radiation sensitivity? It would also be useful to see plots of the FSC between the fitted model and the map. Do you think there may be any overfitting of the model at this resolution?

2) In the Discussion, the authors hint that further improvements in the reconstruction may still be possible from this data set using 3D classification approaches. In these times of rapid progress, it would not be good to publish a structure based on a data set, and then later obtain an improved map from the very same data set. It has been observed in multiple studies lately that 3D classification is a powerful tool to select the best particles and thereby improve resolution of cryo-EM maps. As resolution is the main point of this paper, the authors should try their best in getting as high a resolution as possible from this data set. Therefore, they should consider to at least try the classification route.

Minor comments:

Some have argued that using beam-tilt to illuminate multiple areas per hole would prohibit high-resolution structure determination. Therefore, it would be interesting to know whether beam-tilt was used to collect the data, or whether only stage movement was used?

What does the following sentence mean? “we discarded particle images with the greatest angular uncertainty”. And also how did the authors “analyz[e] the probability distributions over all orientations”? In RELION, those distributions are never written out for the user to assess. I suspect the authors applied a cutoff based on the height of the probability distributions at their maximum (rlnMaxValueProbDistribution)? This should be made clearer, and perhaps the authors could describe how the cutoff was chosen.

Many complexes show local variations in resolution. I guess the 20S proteasome is rather rigid, but it would be good to have a local-resolution map (from bsoft or resmap) anyway.

Others have observed relatively poor densities for negatively charged residues and have blamed radiation damage. How does this look in this higher resolution map? And did these side chains get better from the total-dose map to the radiation-damaged-weighted map after polishing?

In the Discussion, it would be interesting to hear the authors' opinion on why they got significantly beyond the resolution limits observed by others. Is it only the high number of asymmetric units, combined with a symmetrical, rigid particle? Or is there something that others could/should change in their procedures?

Given the uniquely high resolution, I would encourage the authors to submit their 197 selected movies to the EMPIAR archive at EMDB, to enable others to learn from these data.

---

## [Author Response]

*1) All 3 reviewers mentioned in some form that it would be good if you could elaborate somewhat on the obtained atomic model*.

The T20S crystal structure (PDB 1YAR) was rigid body fitted into the cryoEM map using UCSF Chimera (22) and then iteratively refined using Rosetta and Coot (18). We used Rosetta tools recently developed for cryoEM (15) to perform fragment rebuilding, torsion angles and cartesian minimization of atomic positions and B factor refinement. At each iteration, the best model was visually inspected in Coot and a few amino acid side chain rotamers were manually adjusted to best fit the density. Water molecules were also manually added using Coot after the first iteration of Rosetta refinement. Rosetta and Coot refinements were performed using a training map corresponding to one of the 2 maps generated by the gold-standard refinement procedure in Relion. The second map (testing map) was used only for calculation of FSC with the atomic model (17). The quality of the final model was analyzed with Molprobity (14). This elaborated description has been added to the manuscript.

Are there any differences in the rotamer conformations between your cryo-EM structure and the 1.9 Å X-ray structure?

Yes, several side chain rotamers exhibit differences from the 1.9 Å X-ray structure. This is observed either in a region with significant backbone rearrangements as well as at sporadic locations.

And at this resolution do you observe similar effects of radiation damage on specific side chain densities as for example in (Allegretti et al., eLife 3, e01963, 2014; Bartesaghi et al., PNAS 111, 11709-14, 2014), who have shown that the majority of carboxylate chains are missing in cryo-EM maps, supposedly due to their high radiation sensitivity?

We also observe poor densities for negatively charged residues, as reported previously by other groups. This comment has now been added to the manuscript along with the appropriate citations.

*It would also be useful to see plots of the FSC between the fitted model and the map*.

We added to Figure 2 the FSC curve computed between the atomic model and the test map. The FSC reaches 0.5 at 2.86 Å resolution, in agreement with the resolution estimated by gold standard refinement in Relion.

Do you think there may be any overfitting of the model at this resolution?

We agree that overfitting is always a risk at any resolution but the use of independent training and test maps, as we used here, attempts to avoid this pitfall. Also the agreement of the model vs. map FSC resolution curves with the Relion curves as described above, helps provide confidence that the overfitting is at least as limited as possible.

*2) In the Discussion, the authors hint that further improvements in the reconstruction may still be possible from this data set using 3D classification approaches. In these times of rapid progress, it would not be good to publish a structure based on a data set, and then later obtain an improved map from the very same data set. It has been observed in multiple studies lately that 3D classification is a powerful tool to select the best particles and thereby improve resolution of cryo-EM maps. As resolution is the main point of this paper, the authors should try their best in getting as high a resolution as possible from this data set. Therefore, they should consider to at least try the classification route*.

According to the results provided by the ResMap software (which we interpret only qualitatively), the T20S structure is rigid in the majority of the core with only a few very slightly flexible loops on the periphery. We thus do not think that 3D classification would lead to significant improvements in the overall resolution. We feel that we have already done the best we can to get to this point and that the time required (both computational and personnel) for further investigation of possible improvements in resolution would delay the paper for many months and severely reduce its impact. We thus respectfully request that the reviewers accept that, while we suspect further improvements may be possible, these cannot be provided in a reasonable timeframe.

Minor comments:

*Some have argued that using beam-tilt to illuminate multiple areas per hole would prohibit high-resolution structure determination. Therefore*, *it would be interesting to know whether beam-tilt was used to collect the data, or whether only stage movement was used?*

As mentioned in the Methods section of the manuscript, only stage movement was used to collect the data. We have now further highlighted this fact in the Discussion. Also only a single image was acquired per hole, and we have also added this point to the Methods section of the manuscript.

*What does the following sentence mean? “we discarded particle images with the greatest angular uncertainty”. And also how did the authors “analyz[e] the probability distributions over all orientations”? In RELION, those distributions are never written out for the user to assess. I suspect the authors applied a cutoff based on the height of the probability distributions at their maximum (rlnMaxValueProbDistribution)? This should be made clearer, and perhaps the authors could describe how the cutoff was chosen*.

We indeed used the MaxValueProbDistribution criterion to choose which particles to discard. We empirically determined a cutoff of 0.07 after previously determining that cutoffs that retained ∼85% of the particles seemed to yield the highest resolution maps, we have added this to the manuscript.

*Many complexes show local variations in resolution. I guess the 20S proteasome is rather rigid, but it would be good to have a local-resolution map (from bsoft or resmap) anyway*.

We analyzed our map using ResMap, which yielded a nearly flat local resolution map (only the most peripheral regions exhibited lower resolution). The estimated resolution for the majority of the map, however, was 2.2 Å. Based off of the features (or lack thereof) in the map, we do not believe that the resolution is this high, and therefore thought it would be misleading to report these results in the main text. We have included the histogram (Figure 5) and colored-by-resolution map below (Figure 6).Author response image 1.Author response image 2.

Others have observed relatively poor densities for negatively charged residues and have blamed radiation damage. How does this look in this higher resolution map? And did these side chains get better from the total-dose map to the radiation-damaged-weighted map after polishing?

We also observe poor densities for negatively charged residues, as reported previously by other groups. This comment has now been added to the manuscript along with the appropriate citations. Particle polishing does not fully restore the densities corresponding to these side chains.

*In the Discussion, it would be interesting to hear the authors' opinion on why they got significantly beyond the resolution limits observed by others*. *Is it only the high number of asymmetric units, combined with a symmetrical, rigid particle? Or is there something that others could/should change in their procedures?*

We believe several factors played a role in obtaining the resolution reported here. These include: (i) sample quality (rigidity, homogeneity); (ii) microscope/camera settings and performance; (iii) the relatively high dose we used compared to several other studies may help to properly align particle images; (iv) the maturity of the algorithms used (projection matching, particle polishing).

An additional paragraph has been added to the Discussion section outlining these factors.

*Given the uniquely high resolution, I would encourage the authors to submit their 197 selected movies to the EMPIAR archive at EMDB, to enable others to learn from these data*.

Thank you, this has been done.